# Skin-on-a-Chip Technology: Microengineering Physiologically Relevant In Vitro Skin Models

**DOI:** 10.3390/pharmaceutics14030682

**Published:** 2022-03-21

**Authors:** Patrícia Zoio, Abel Oliva

**Affiliations:** 1Instituto de Tecnologia Química e Biológica (ITQB), Universidade Nova de Lisboa, Avenida da República, Estação Agronómica Nacional, 2780-157 Oeiras, Portugal; patricia.zoio@itqb.unl.pt; 2Instituto de Biologia Experimental e Tecnológica (IBET), 2781-901 Oeiras, Portugal

**Keywords:** reconstructed skin models, tissue engineering, organ-on-a-chip, microfluidics, dynamic culture, drug testing, dermal absorption

## Abstract

The increased demand for physiologically relevant in vitro human skin models for testing pharmaceutical drugs has led to significant advancements in skin engineering. One of the most promising approaches is the use of in vitro microfluidic systems to generate advanced skin models, commonly known as skin-on-a-chip (SoC) devices. These devices allow the simulation of key mechanical, functional and structural features of the human skin, better mimicking the native microenvironment. Importantly, contrary to conventional cell culture techniques, SoC devices can perfuse the skin tissue, either by the inclusion of perfusable lumens or by the use of microfluidic channels acting as engineered vasculature. Moreover, integrating sensors on the SoC device allows real-time, non-destructive monitoring of skin function and the effect of topically and systemically applied drugs. In this Review, the major challenges and key prerequisites for the creation of physiologically relevant SoC devices for drug testing are considered. Technical (e.g., SoC fabrication and sensor integration) and biological (e.g., cell sourcing and scaffold materials) aspects are discussed. Recent advancements in SoC devices are here presented, and their main achievements and drawbacks are compared and discussed. Finally, this review highlights the current challenges that need to be overcome for the clinical translation of SoC devices.

## 1. Introduction

Skin and subcutaneous disorders affect approximately one-third of the global population and are associated with a burden encompassing psychological, social and financial dimensions [1]. In particular, chronic skin diseases, such as psoriasis, eczema and atopic dermatitis, result in significant morbidity and affect patient quality of life [2]. On the other hand, malignant skin diseases, such as malignant melanoma, are frequently fatal [3]. The high prevalence of skin diseases combined with the emergence of new technological advancements in drug development is fueling the growth of the skin-targeted drug delivery market.

Skin-targeted drug delivery systems include topical, dermal and transdermal approaches. In topical drug delivery, the active substances are intended to remain on the skin’s surface (e.g., barrier creams, sunscreens and repellents) whereas dermal delivery targets active substances into the relevant skin layers (e.g., corticosteroids and antibiotics). In parallel, transdermal drug delivery research has also seen an upsurge in recent years. This approach could circumvent the complications of oral and intramuscular drug delivery and achieve controlled systemic delivery of drugs. Recent innovations in this field include a transdermal patch indicated against symptoms of Parkinson’s disease [4] and a skin-targeted vaccine against COVID-19 [5].

The successful development of skin-targeted drug delivery systems requires careful consideration of the human skin’s anatomy, physiology and physicochemical properties. This evaluation is important to achieve the desired drug effect and avoid adverse reactions such as skin sensitization and/or irritation. The development of new pharmaceutical drugs would greatly benefit from biomimetic in vitro skin models that could replicate the key components of the in vivo healthy and diseased human skin.

Conventional preclinical drug testing relies on in vitro cell cultures and animal models. Most commonly, in vitro cell culture relies on two-dimensional (2D) cell culture systems, typically monolayers of epidermal keratinocytes and/or dermal fibroblasts. While these models offer a rapid, reproducible system to study drug responses, they are not good predictors of the complex interactions seen in vivo [6]. The lack of a 3D physiological tissue environment greatly minimizes the models’ physiological relevance and applicability. On the other hand, in vivo animal models offer information on systemic effects but cannot replicate human skin anatomy and physiology. During the development of pharmaceutical skin-targeted formulations, mouse models are often mandatory for in vivo translational research. However, mouse skin is structurally and functionally different from human skin; it is thinner, contains more hair follicles, includes fewer keratinocyte layers, presents decreased barrier function and greater absorption [7]. Moreover, animal models suffer from low throughput and interspecies variability [8]. These flaws in the conventional testing methods result in a lack of correlation between the input (drug candidates) and output (approved drugs), contributing to the R&D decline [9].

From an ethical perspective, the replacement of animal models satisfies a growing societal concern regarding animal experimentation. Ethical guidelines dictate that, where possible, animal experimentation should be replaced, reduced, or refined (3R principle). The cosmetic industry has been greatly affected by the restrictions imposed on animal testing. Since 2009, the European Commission has been approving regulations on cosmetics, establishing a testing and marketing ban: a prohibition against testing finished cosmetic products or ingredients on animals and commercializing any cosmetic product or ingredient that has been tested on animals within the European Union [10]. The European Centre for the Validation of Alternative Methods (ECVAM) was established in 2010 as a reference laboratory for researching and validating alternative methods, following 3R principles [11]. In 2013, a full marketing ban was put in place for all human health effects tested in animals, including repeated-dose toxicity, reproductive toxicity and toxicokinetics, irrespective of the availability of alternative non-animal tests.

The combinatory effect of the high prevalence of skin diseases, R&D decline and restrictions on animal testing pressured the development of physiologically relevant skin models that could replace conventional, inefficient approaches. Recently, 3D cell culturing techniques have improved the relevance of the available models and demonstrated the synergistic effects that different cell types have on each other [12]. These models can be assembled into complex structures to simulate more physiologically relevant conditions. Both reconstructed human epidermis (RHEm) and full-thickness skin models (FTSm) have been used for many applications including basic, pharmacological and cosmetic research. Innovative techniques such as 3D printing and scaffolds are promising approaches to increase the relevance of these models. However, current tissue-engineered skin models still fall short of the desired controllability and are deficient in several essential key components of the in vivo skin. In particular, their lack of vascularization results in restricted nutrition supply, cell–cell and cell–extracellular matrix (ECM) interactions.

The need for physiologically relevant and functional tissue models led to new technologies for cell cultures such as organ-on-a-chip (OoC) or microphysiological systems. This modern technology aims to surpass the limitations of the 3D cell-based culture platforms and increase the predictive power of in vitro models (Figure 1) [13]. These systems have the potential to achieve experimental controllability and reproducibility similar to 2D cell culture while allowing for increased physiological relevance and complexity. In the last few years, OoC technology has been used to recreate advanced biomimetic skin models, known as skin-on-a-chip (SoC) models [14]. In this Review, key technological and biological aspects for developing physiologically relevant SoC models will be discussed. The state-of-the-art SoC devices will be presented, and the major advancements and drawbacks will be highlighted.

## 2. Organ-on-a-Chip Technology

OoC technology combines advancements in tissue engineering and microfluidics to reproduce critical functional aspects of human tissues and organs [15]. The key features of OoC platforms include the presence of biomechanical forces and the integration of multiple cell types to model their complex interactions [16]. Perfusion channels are included on the OoCs to model fluid flow across the tissues. These channels act as engineered vasculature, delivering cell culture media to the cells within the culture chamber and removing associated cell metabolites and detritus. By growing tissues inside a controlled environment and mimicking in vivo-like forces, it has been possible to create more advanced models better suited for pharmaceutical applications and disease modeling. This technology can substantially reduce the total costs of drug screening due to the decrease in reagent volumes and the possibility of testing various drug candidates in parallel [9]. Additionally, the close-loop environment in OoC makes possible the integration of various biosensors for real-time monitoring of tissue function [17]. The sensors can be useful for monitoring the formation of healthy tissues and monitoring disease conditions in response to drug candidates.

The potential of OoC platforms has been extensively demonstrated in the last decade. It was shown that the stimuli applied on the chip lead to alterations in cell behavior, including improved cell morphology and differentiation and more in vivo-like interactions between cells and the ECM [18]. Some important examples include the reconstruction of a lung-on-a-chip capable of recreating relevant in vivo mechanical forces (stretching) [19], a liver-on-a-chip mimicking parenchymal hepatocytes and the sinusoidal space [20] and a cardiac muscle grown on chip mimicking topographical and electrical cues, important for achieving alignment and maturation in cardiac tissue [21]. Other relevant models produced using OoC technology include bone, blood-brain barrier, skeletal muscle, eye, gut and spleen [22]. Additionally, these platforms have been used to recreate multiple pathological conditions, important for future evaluation of drug candidates and therapies [23]. The failure in drug development for cancer treatment is, in part, a result of conventional cancer models failing to recapitulate the in vivo tumor microenvironment. Despite its complexity, various groups are successfully using OoC technology to study cancer’s basic cellular and molecular biology. OoC can be an important tool for future drug development in this field. Currently, important models have been developed to model the tumor microenvironment, cancer cells’ trans-endothelial migration and cancer-related angiogenesis [24].

OoC technology is a promising tool to increase the predictive ability of skin models, offering continuous replenishment of oxygen, nutrients and mechanical stimulus.

## 3. Key Requirements for the Development of Skin-on-a-Chip Devices

A physiologically relevant SoC model is expected to include the main layers of the human skin (dermis and epidermis) and a vascular system. The cell source and scaffold type are crucial to obtain a final model with the desired architecture and physiology. The integration of mechanical stimuli such as cyclic stretching and shear stress should also be considered to reproduce the in vivo-like microenvironment. Finally, the integration of sensors in the SoC should be considered for real-time monitoring of skin function. Figure 2 gives an overview of the main considerations when developing a SoC model.

### 3.1. Cell Sourcing

One of the determinant factors for developing a SoC device is cell selection and sourcing. When choosing the ideal cell types for integration in the SoC platform, the context of use needs to be considered and the key aspects and components needed for function identified.

Multiple *in-house* culture protocols have been developed and optimized over the last decades to incorporate different cell types and biological sources on human skin models. Conventional FTSms are constructed using primary human skin cells. These cells are isolated from healthy human skin obtained from standard surgical procedures. The clear advantage of using primary cells from human donors is capturing the in vivo phenotype [25]. The most common FTSms consist of an epidermis generated from primary keratinocytes and a dermal compartment populated by primary fibroblasts. These models allow cross-talk between keratinocytes and fibroblasts, increasing physiological relevance compared to RHEms. More complex in house models report adding other cell types into the epidermal (e.g., melanocytes and Langerhans cells) or the dermal compartment (e.g., fibroblasts and lymphocytes) [26]. Zoio et al. generated a pigmented FTSm for long-term studies using primary fibroblasts, keratinocytes and melanocytes [27]. The group reported the formation of a mature dermis and a terminally differentiated epidermis that could maintain its architecture for up to 50 days. Primary cells can also be successfully used to model pathologies when sourced from donors with the disease being studied. For example, in vitro RHEms displaying characteristics of psoriatic epidermis were successfully developed using keratinocytes from patients with psoriasis [28]. Recently, Rioux et al. successfully generated a 3D psoriatic skin model using patient-derived skin cells [29]. Moreover, activated T cells, isolated from human whole blood, were incorporated to develop an immunocompetent model reflecting the psoriatic inflammatory environment. Overall, primary cells present a unique set of advantages. However, these cells also include important drawbacks such as the limited availability of donor skin, donor variation that may hamper experimental readout parameters and restricted proliferation and amplification capacity.

As an alternative, cell lines with validated purity and viability can be considered. The main advantages of cell lines are their reproducibility and the high availability of reliable protocols for cell expansion. However, cells lines are only approximations of the primary cell function and can deviate from the original phenotype. This is the case of the widely used keratinocyte cell line, HaCaT, extensively investigated for its ability to undergo differentiation. Although epidermal tissue formation is possible using HaCaT cells, its low differentiation potential compared to primary keratinocytes makes it difficult to generate a functional *stratum corneum* [30]. Therefore, the use of the HaCaT keratinocyte cell line is unsuitable for the development of physiologically relevant SoC models as it fails to develop a fully stratified epidermis [31].

Alternatively, hTERT-immortalized keratinocyte cell lines have been proved to develop high-quality epidermal and skin equivalents. These cells faithfully mimic primary keratinocyte proliferation, differentiation, skin barrier function and inflammatory responses. Reijnders et al. developed a fully differentiated FTSm generated from hTERT-immortalized keratinocytes and fibroblasts [32]. The group reported the successful production of a FTSm with a fully differentiated epidermis and a fibroblast-populated dermis comparable to a FTSm originating from primary cells. Models including hTERT-immortalized cells can provide higher throughput screening of new products and drugs and should be considered for incorporating into reproducible SoC platforms. However, depending on the context of use, donor variation can be an important feature to resemble the population accurately. Cell lines lack patient-specificity, which is particularly relevant for disease modeling studies. Furthermore, studies point to cell lines having induced overexpression of proteins involved in toxicity-related pathways [33]. Therefore, these cells can limit the use of the models for toxicity testing.

Induced pluripotent stem cells (iPSC) are a promising cell source for tissue engineering. These cells are derived from adult somatic cells via reprogramming with ectopic expression factors (Oct3/4, Sox2, c-Myc and Klf4) [34]. The expression of these factors results in the suppression of genes responsible for differentiation, reverting the cells to a pluripotent state. The use of iPSC could circumvent the current limitations of FTSms due to their unlimited growth and ability to differentiate into multiple cell types. Furthermore, cells differentiated from iPSC retain characteristics of the original donor, such as disease phenotype, offering an alternative source for modeling skin diseases [35]. Gledhill et al. generated a functional FTSm from iPSC-derived keratinocytes, melanocytes and fibroblasts containing a functional epidermal-melanin unit [36]. This model showed similar morphology and functionality to a healthy skin model. However, many factors need to be addressed before iPSC-derived FTSm can be a viable option. These factors include the high cost, retention of epigenetic memory and genomic instability.

Finally, the cell source could affect how the cells are able to sense the mechanical forces acting on them. Sivarapatna et al. showed differences in response to shear stress in microvessels generated using iPSC but not in microvessels generated using primary human umbilical vein endothelial cells (HUVECs) [37]. The group concluded that iPSC-derived endothelial cells are more plastic in modulating their phenotype under flow than HUVECs. The different cell plasticity in response to shear stress could be a determinant factor in the production of a physiologically relevant SoC model. Overall, the development of a physiologically relevant SoC device requires an understanding of the impact of cell source and type on the models’ architecture and physiology. Each cell type has its own set of advantages and drawbacks that will need to be considered according to the specific application.

### 3.2. Cell Scaffolds

One of the major challenges for producing a biomimetic skin model is the development of a dermal matrix with in vivo-like composition and mechanical environment while assuring its stability and reproducibility. The in vivo dermis is an integrated system of fibrous, filamentous and amorphous connective tissue, responsible for the elasticity and tensile strength of the skin. The dermal fibroblasts are surrounded by the ECM composed of various proteins, including collagens, glycosaminoglycans, proteoglycans and adhesive proteins such as fibronectin and laminin (Figure 3a) [38].

The dermal components are secreted and reorganized by the fibroblasts, resulting in an organized meshwork that provides a soft and elastic environment for cellular growth and interactions. Dermal fibroblasts communicate extensively with the keratinocytes within the epidermis, located on top of this elastic dermal layer. The top of the dermal layer acts as a soft substrate for keratinocyte adherence and the formation of an extracellular basement membrane. The correct development of a SoC model requires an understanding of the interplay between different cell types and the effect of the ECM on cellular function and architecture. The ECM composition and 3D arrangement affect cell morphology, polarity and survival and must be carefully engineered to promote the formation of appropriate tissue architecture and physiology [39].

Conventional methods to develop the dermal compartment involve using natural hydrogels, typically animal-derived collagen, to create an environment conducive to cell adhesion and growth. However, as fibroblasts proliferate, the ECM contracts, limiting their lifespan and application [40]. This also frequently results in a reduction of volume and detachment of the compartment from the support membrane leading to leakage problems. The use of conventional hydrogel materials constitutes a relevant limitation for the standardization of FTSm and their reliable application, making them unsuitable for integration on SoC devices. To overcome these problems, various groups proposed chemical and physical modifications of the matrix. Lotz et al. were able to generate cross-linked collagen hydrogels that were more mechanically stable and less prone to enzymatic degradation when compared with non-cross-linked collagen hydrogels [41]. The produced dermal compartments could support a fully differentiated epidermis on top. Overall, these stable structures are more suitable for integrating a SoC device. However, they typically comprise of one ECM component (e.g., collagen type I) which does not represent the complex in vivo dermal composition. Alternative polymers or hydrogels should be considered to recreate the biophysical properties of the skin. One possibility is to generate a fibroblast-derived matrix in which fibroblasts are stimulated to synthesize the different components of the in vivo ECM [42]. This strategy can be combined with a scaffold of high porosity to achieve good mechanical stability [27,43].

Combining an in vivo-like ECM with a perfusion system makes it is possible to recreate physiological interstitial flow. This flow refers to the fluid movement through the ECM or *interstitium* of a tissue and constitutes an essential component of microcirculation (Figure 3b). These forces provide the convection necessary for the transport of molecules and provide a specific mechanical environment to cells [44]. A biomimetic skin model should recreate interstitial flow. This is possible by generating a pressure gradient across the SoC through fluidic flow in the basal chamber, especially during culture at ALI, which increases interstitial flow in the FTSm. This enhances the transport of nutrients and induces morphogenic effects in both fibroblasts and keratinocytes. In particular, in vitro studies showed that interstitial flow affects human fibroblasts’ alignment and differentiation [45]. A careful consideration of the interstitial flow and how it is sensed by cells to drive morphogenic responses and other biological responses is relevant for developing a SoC platform, with important implications on the delivery of drugs and therapeutics.

### 3.3. Vascularization

The vasculature of the skin is located within the dermis and serves essential functions, delivering nutrients to the cells and removing unwanted metabolic waste products. Furthermore, it acts as a conduit for elements of the immune system and plays a role in thermoregulation [46]. The cutaneous vasculature is also involved in various pathological conditions, including acute and chronic inflammatory conditions, tumor growth, metastasis of malignant melanoma and wound healing [47,48,49]. Moreover, it has been shown that the vasculature impacts the transdermal penetration of substances and the response to irritants [50]. Therefore, the generation of cutaneous vasculature is important to model physiological in vitro skin and pathophysiological conditions.

Conventional reconstructed skin models do not include vascularization, which limits their ability to reflect the function of in vivo human skin. Moreover, the lack of vascularization greatly reduces the supply of oxygen and nutrients in 3D skin tissues, reducing cell viability in thick (millimeter-sized) skin constructs [51]. In particular, the diffusion limit of oxygen in cell-rich tissues is ~200 µm. This value is used to determine the smallest cubic voxel of cells that can survive without vasculature (functional unit). Therefore, the culture of thicker skin constructs (>200 µm) can undergo hypoxia and apoptosis. The importance of vascularization, in particular for implanted skin grafts, pressured the development of innovative approaches to recreate in vivo-like vasculature. One popular strategy to generate vascularization in skin grafts relies on the host vessels’ ability to invade the implanted engineered tissue [52]. Alternative approaches to induce vasculogenesis in vitro include cell seeding onto scaffolds or hydrogels, cell sheeting engineering and cell encapsulation [53]. However, most of these approaches involve complex and slow processes to induce vascularization, not suited for high-throughput applications. Furthermore, the vascular channels formed in these studies have a random formation and are inaccessible to external pumps. Consequently, these models do not have functional perfusable vascular channels, limiting their applicability.

Advances in microfluidics, 3D printing and micro-molding have enabled the development of perfusable vascular networks [54]. Prevascularized patterning methods used to create vascular-like networks on-chip can be divided into soft lithography techniques and 3D patterning methods. Soft lithography typically results in square or rectangular cross-sectioned channels, which do not represent the in vivo 3D geometry. To reproduce 3D barrier tissues on-chip, the most common approach is the integration of a porous membrane between two-channel, polydimethylsiloxane (PDMS) layers (Figure 4a).

However, membrane-based models prevent direct cell-to-cell interaction between the vascular and parenchymal components. Alternatively, microfabrication techniques can be adapted to embed hydrogels in the PDMS device (Figure 4b). These devices allow direct contact between the vascular component and the parenchyma. Furthermore, vasculogenesis can be induced in these models to create a vascular network in the central hydrogel.

Sophisticated 3D patterning methods can be used to reproduce more physiologically representative vascular networks on-chip. These methods include templating techniques, layer-by-layer manufacturing and 3D bioprinting. Templating techniques are a subtractive strategy in which a material with a specific geometry is embedded in a hydrogel and subsequently removed/dissolved (Figure 4c). Although templating offers a simple method to reproduce hollow channels on a matrix, it is typically limited to simple geometries. Moreover, this technique is unable to reproduce dimensions relevant to model capillaries as seen in skin models fabricated angiogenically. Layer-by-layer techniques could be used to generate more versatile perfusable vascular networks via multilayer assembling (Figure 4d). This bottom-up technique consists of assembling 2D prepatterned gel layers into multi-layered 3D geometries. Finally, combining 3D printing with microfluidics could generate complex structures mimicking the native skin due to its precision and versatility (Figure 4e). Three-dimensional printing facilitates the precise extrusion and localization of multiple cell types and biomaterials. Importantly, the bioengineered vascular network should be perfused to achieve adequate tissue function. For more information regarding in vitro strategies to vascularize 3D models, we refer the reader to a more detailed review [54].

### 3.4. Mechanical Stimulus

OoC technology can be used to introduce mechanical stimulus similar to what organs and tissues experience in vivo and, simultaneously, continuously deliver cell culture media and removal of cell metabolites. Cellular mechanotransduction is a well-known phenomenon, with various studies describing the cellular mechanisms where mechanical forces are transduced into biochemical signals [55]. These responses are integral parts of the cellular microenvironment that modulate various processes in health and disease states. Relevant in vivo-like biomechanical forces include shear stress, tensile and compressive forces (Figure 5). Shear stress refers to the force created when a tangential force acts on an object. This force can be imparted by fluid flowing over an object and is prevalent throughout the human body. This effect is more pronounced on the endothelial cells, which are sensitive to changes in fluid flow [56].

Agarwall et al. showed that epidermal keratinocytes are sensitive to microflow-induced shear stress [57]. These cells exhibited mechanoresponsive structural reorganization under the influence of shear stresses of magnitude 0.06 dyne/cm^2^ and cellular damage under shear stresses of magnitude 6 dyne/cm^2^. The group also showed that shear stress increased the cell colony area and expression of E-cadherin and Zonula occludens-1 at the cell–cell junction. These observations point to an improvement in the epithelial phenotype. However, the positive effect of the shear stress on keratinocytes is not expected to play a major role in 3D FTSms developed on-chip. Skin differentiation requires culture at the air–liquid interface, which means that keratinocytes are not exposed to fluid flow induced shear stress at the apical side, except for an initial period of approximately 24–48 h. At the basal side, keratinocytes are typically protected from the effect of shear by a membrane or dermal compartment.

Studies also have shown that shear stress affects fibroblasts’ behavior including their orientation, migration and adhesion [58]. However, in an in vivo-like engineered dermal compartment, the ECM matrix’s pores impede the fluid’s momentum transport outside the individual pores. Therefore, shear stresses acting on dermal fibroblasts are often negligible. Arguably, in a 3D SoC model with vascularization, the most important role of fluid-induced shear stress is on endothelial cells. Several studies described the importance of endothelial cell sensitivity to shear stress and the involvement of these forces on multiple physiological vascular processes [59]. In particular, Tsvirkun et al. developed a microvasculature-on-chip and showed that endothelial cells submitted to physiological levels of flow-induced shear stress demonstrate strong similarities with in vivo capillaries [60].

Various approaches have been explored for perfusing culture media, categorized into pump- or gravity-driven solutions [61]. The most common methods for generating fluid flow are external pumps, including a syringe pump or peristaltic pump. These systems deliver accurate, fine-tuned fluid flow but are typically time-consuming. Furthermore, tubing and connections can increase the risk of contamination. To overcome some of the limitations of external pumping, pumps have been integrated within microfluidic chips to miniaturize the systems. While internal on-chip pumping makes it possible to reduce the setup footprint, integrating these systems is typically complicated regarding fabrication and operation. Alternatively, passive flow, typically gravity driven, has recently emerged as an alternative [62]. For this, rocking platforms are custom-built to recirculate culture media through the microfluidic channels. These approaches exclude tubing and complex setups with a big footprint. However, these methods typically lack fine control of flow rate, are prone to changes in hydrostatic pressure over time and do not include fresh media perfusion and waste removal.

In addition to the shear stress induced by perfusion, tissues in the body are continuously subjected to other forces, including tensile and compressive forces. The tensile forces applied to the cellular microenvironment result in tissue stretching. The human skin is exposed to various types of mechanical stimuli including cyclic stretch. This is a result of environmental effects, growth and various internal processes. Various experiments investigated the effects of stretch stimuli on 2D skin cells including epidermal keratinocytes and dermal fibroblasts. According to the reports, applying tensile forces to dermal fibroblasts triggers signal transduction from ECM into cells, resulting in increased synthesis of collagen I, collagen III and elastin [63]. Furthermore, increased production of protease inhibitors such as tissue inhibitor of metalloproteinase was reported [64]. In epidermal keratinocytes, stretch stimulation promoted cellular adhesiveness, proliferation and protein synthesis [65].

Given the complex interactions between keratinocytes and fibroblasts, other groups have studied the importance of these forces and the effect on skin models containing both cells. Lü et al. developed an in vitro tensile device to study the mechanisms regulating keratinocyte migration under application of mechanical stretch [66]. Their results showed that, when exposed to these forces, dermal fibroblasts increase secretion of epidermal growth factor, promoting an asymmetric keratinocyte migration. This phenomenon results in accelerated wound repair. Tokuyama et al. designed an experimental system wherein stretch was applied during the formation of an FTSm [65]. The group analyzed the effects of these stimuli on keratinization of epidermal keratinocytes and formation of a basement membrane. Cyclic stretch resulted in an increase in the synthesis of basement membrane proteins and a thicker epidermal layer.

Various actuation approaches have been developed to induce cyclic stretch on cells [67]. Researchers either use their stretching platforms, typically employing pneumatic approaches, or use commercially available systems [Flexcell (Burlington, NC, USA), StretxCell (San Diego, CA, USA)]. Although multiple solutions exist to apply cyclic stretch, OoC technology presents a unique opportunity to create customizable solutions and combine them with other relevant forces such as shear stress. Various OoC platforms have reproduced different types of mechanical strains including cyclic stretch. Huh et al. reported a lung-on-a-chip made of a central chamber divided by an elastic porous membrane that could be stretched unidirectionally by providing a vacuum in two adjacent chambers [68,69]. This platform was used to study inflammatory responses and recreate drug toxicity-induced pulmonary edema. Other groups improved on this model by generating a 3D mechanical strain, better simulating the in vivo forces [70].

### 3.5. Design and Fabrication

For FTSm generation, the apical chamber must ensure the correct formation of stratified epithelium and air circulation. The lower chamber must ensure correct medium perfusion. Typically, at the end of an experiment, the tissue should be retrieved to perform an analysis of its architecture and physiology. Conventional methods for developing biological barriers, including skin, typically involve using permeable supports (e.g., transwell), establishing two chambers that separate the apical and basal surfaces of the tissue. This correct compartmentalization is crucial for forming 3D skin models with an in vivo-like barrier. It allows a leakage-free separation between the apical and basal surfaces of the skin, allowing the correct establishment of the ALI.

Advances in biomaterials and microfabrication have allowed researchers to integrate compartmentalization inside OoC platforms to develop innovative barrier models. The most common approach for creating separated domains inside an OoC is integrating a semipermeable membrane between two elastomeric microfluidic channels (Figure 6). This model was first envisioned by Huh et al. to reproduce the ALI of the lung [71]. Since then, this approach has been used to recreate multiple tissues, including the blood-brain barrier, gut and liver [72]. Important limitations of this technique were briefly mentioned in Section 3.2 and included the lack of direct interface between the vascular channels and the parenchyma, limiting the generation of in vivo-like vascularization. The use of membranes separating the tissue from the fluidic compartment limits mass transport and reduces the prediction of transdermal skin absorption into the circulatory system. Moreover, this approach typically relies on the irreversible bonding between the apical and basal culture environments, using adhesive glues or oxygen plasma treatment [73]. For the generation of FTSms, this strategy presents several limitations, including difficulties related to the introduction of a dermal matrix on top of the membrane, culture maintenance and challenges accessing the tissue for end-point analysis.

A fully realized SoC model with the potential to replace animal experiments needs to replicate the structure and function of the in vivo skin at a relatively low cost and be compatible with high-throughput studies. In the academic field, most OoCs are fabricated from PDMS using soft lithography techniques. This material has been commonly used due to its biocompatibility, gas permeability and rapid prototyping capabilities. However, PDMS is well known for its tendency to absorb small hydrophobic molecules and other organic compounds and drugs, limiting its applicability [74]. Moreover, large-scale production and commercialization of PDMS-based devices have been a challenge due to the high cost of the current fabrication strategies. Novel low-cost methods for the fabrication of PDMS devices, including mask-free procedures and alternatives to plasma treatment, could expand their applicability and commercialization [75].

Thermoplastics are gaining more attention since they offer several advantages compared to PDMS: they are cheaper, present less absorption and evaporation and provide higher chemical resistivity. The use of thermoplastics can lead to an easier transition from academia to industry since they can be fabricated both by rapid prototyping and industrial-scale fabrication techniques. Integration of membranes on thermoplastics can be performed using various techniques such as thermal bonding, ultrasonic welding, laser welding, solvent bonding and surface functionalization [76]. Alternatively, these materials can be reversible sealed using clamps or magnets, providing minimal requirements for membrane materials. This technique is compatible with the development of modular microfluidic circuits, a concept several research groups have explored. Abhyankar et al. developed a module-based approach in which magnetic latching allows culture membranes to be sandwiched between fluidic channels and sealed in place to support compartmentalized cultures [77]. The flexibility of this design could be interesting for a future SoC, allowing easy removal of the skin tissue for histological and immunohistochemistry analysis and allowing direct cell seeding and introduction of the hydrogel mixture for dermis formation onto the membrane. Furthermore, different modules could be developed, enabling the user to select the modules that meet their requirements. For example, a dedicated apical module with integrated microneedles could be developed to study the effect of these devices on the delivery of pharmacologically active ingredients. Microneedle devices are increasing in popularity due to their potential for transdermal delivery, disease treatment and diagnosis [78,79]. Finally, a user-friendly interface is also an important consideration when designing a SoC device. This could streamline experimental procedures and encourage adoption in laboratories unfamiliar with OoC technology.

Experimental studies can be complemented with numerical simulations to assess the viability of the SoC devices and optimize their design. These tools can be used to predict critical parameters such as oxygen concentration, fluid velocity, shear stress and diffusion processes. Consequently, it is possible to better determine the required chip materials, chamber/channel geometry and dimensions, fluid properties and testing conditions to develop physiologically relevant models.

Hernando et al. showed the importance of performing in silico studies for design optimization by modeling cell behaviors in OoC devices with different channel geometries. The group reported that the cell culture time required to fully exploit their OoC could be reduced by redesigning the chip’s inlet channel and chamber network [80]. Zahorodny-Burke et al. used numerical simulations to evaluate the impact of the chip’s materials on the oxygen concentration in cell culture. The group reported that OoC devices fabricated using PMMA and cyclic olefin copolymer (COC) resulted in a low oxygen supply to the cells. They concluded that flow rates should be optimized to increase oxygen supply when using materials with low oxygen diffusion [81]. More recently, Kheiri et al. used computational modeling to simulate multiple device designs and flow conditions for reproducing tumor spheroid-on-a-chip [82]. The group concluded that computational modeling is an efficient strategy to optimize microfluidic device designs, providing insightful information on the drug transport phenomena in these models.

Similarly, numerical simulations can be used to simulate the microenvironments of SoC devices, towards developing models of greater accuracy. The adequate simulation of flow and drug transport in these models requires an understanding of the key physical properties of the skin. Narasimhan et al. described a detailed numerical model for transdermal drug delivery by considering the skin as a composite, porous material [83]. The authors provided the equations governing the diffusion through the porous medium in the different skin layers. A detailed description of mathematical models that can be used to predict fluid movement and drug diffusion across the SoC models is also provided in a recent publication by Ponmozhi et al. [84].

### 3.6. Sensor Integration

Conventional analyses of tissue function mainly depend on endpoint assessment techniques. In particular, for 3D cell culture using scaffolds or membranes, the monitoring cannot be done using conventional microscopy techniques due to the limited light penetration and scattering effects [85]. The conventional process usually includes the removal of the tissue from the wells, chemical fixing, and labelling. Alternatively, assays based on the transport of tracer compounds (e.g., fluorescein isothiocyanate-labelled dextran) are performed. However, these compounds can affect the tissue’s integrity and lack the sensitivity to detect subtle changes in their function [86]. With OoC systems, it is possible to surpass this limitation by integrating microsensors for measuring relevant physical/chemical parameters in situ.

Integrated sensors can characterize the engineered tissue and tissue interactions with different stimulants, giving prompt insight. First attempts to integrate sensors inside the OoC systems have shown the relevance of monitoring in real-time to capture changes in the properties of living cells to obtain complete information about what is happening inside the OoC. An example is the work of Zhang et al. of an automated multi-tissue organ system that integrates an array of on-chip sensors, including electrochemically activated immune biosensors attached to physical microelectrodes, mini microscopes and optical pH, oxygen and temperature monitors [87]. This technical feat highlights the ongoing engineering advances that are enabling real-time non-invasive monitoring of OoC microenvironments.

In the context of SoC platforms, the integration of sensors would be extremely relevant. Growing a FTSm inside a platform is a very long process ranging from 2 weeks to 1.5 months. Using conventional endpoint assays, no information is acquired during the period necessary for skin formation, resulting in low experiment reproducibility. An ideal future SoC platform should integrate physical sensors for monitoring relevant cell culture parameters (e.g., pH, O2, temperature), electrochemical sensors for measuring soluble protein biomarkers as well as transepithelial electrical resistance (TEER) sensors to measure the skin barrier function.

#### TEER: Applicability and Challenges

TEER measurements enable non-destructive, real-time quantification of the barrier function. This technique accesses the integrity of stratifying cultures by measuring changes in the transcellular and paracellular permeability of epithelial/epidermal in vitro cell cultures. TEER measurements have become a widely accepted method to evaluate the tissue barrier function in vitro systems including blood-brain barrier, gastrointestinal and pulmonary models [88]. This technique is also showing promising results in the context of skin engineering.

Multiple studies have reported the use of TEER to follow FTSm/RHEm formation during culture, reflecting the formation of a functional barrier [27,89,90,91]. TEER values increase gradually during skin culture, which correlates with the differentiation process. This increase in TEER reflects the formation of tight junctions in the vital cell layer and *stratum corneum* [92]. In addition to monitoring epidermal differentiation, TEER measurements could be used for drug testing. In fact, TEER measurements are a testing parameter of the Organisation for Economic Co-operation and Development (OECD) test guidelines 430 (In vitro skin corrosion: Transcutaneous electrical resistance Test Method). Multiple studies explored the potential of TEER measurements to evaluate the skin irritation potential of different compounds. Wei et al. tested 46 compounds for their skin irritation potential on RHEms and FTSms using a combination of TEER, tissue viability and cytokine measurements [93]. The group reported TEER as a more sensitive endpoint of skin irritation than tissue viability measurements. Abdayem et al. used TEER to compare the effects of two excipients on RHEm, applied topically or “systemically”, that is by addition to the culture medium [94]. Groeber et al. used TEER as a complementary endpoint in skin toxicity testing to allow distinguishing between the effect of strong irritants and non-irritants [91]. The group concluded that TEER could be particularly useful to identify sub-irritative effects on the skin such as stinging, burning and itching sensations. Similarly, Zoio et al. reported TEER as a useful technique to identify the effect of a test substance on the outer epidermal layers, such as the *stratum corneum* [95]. Furthermore, the group also used TEER measurements for quality control by evaluating the integrity of the barrier function prior to their use in the subsequent assays.

Taken together, these studies demonstrate the potential of TEER measurements for the quantitative measurement of skin barrier function. This parameter has been used to evaluate the stages of skin differentiation in real-time and the impact of topically or systemically applied substances. However, the literature has reported a wide range of TEER values for the same cell/tissue type [88]. It has been hypothesized that this high variability results from a combination of biological factors (e.g., cell passage number) and TEER-related mechanoelectronics. Conventional TEER measurements are typically performed using commercially available tetrapolar (four-electrodes) configurations. In these systems, the current is injected through a pair of electrodes, current-passing (CP), and another pair of electrodes, voltage-sensing (VS), measures the voltage drop. For this, an alternating current (AC) with a typical frequency of 12.5 Hz is applied using a commercially available voltOhmeter (e.g., EVOM/World Precision Instruments/FL/USA). This frequency is used to avoid charging effects on the electrodes and cell layer. The most commonly used tetrapolar electrodes are the commercially available chopstick/STX2 electrodes, which are placed at both sides of a cellular barrier (Figure 7a). However, the use of handheld chopstick electrodes can induce variability between measurements due to variations in depth and/or angle of immersion. Moreover, these sensors cannot deliver a uniform current density when large tissue culture inserts are used (>12 mm in diameter) [96,97]. Alternatively, an electrode chamber (e.g., EndOhm chamber) containing a pair of concentric electrodes can generate a more uniform current density across the tissue when compared to the chopstick electrodes (Figure 7b). Moreover, the Endohm’s fixed geometry increases reproducibility.

TEER measurements can also be performed on OoC devices by introducing electrodes on both sides of the cellular barrier. However, this can be challenging due to the closed, micrometer-sized channels in OoC [98]. TEER measurements on-chip can be achieved by inserting electrodes on the chip inlets/outlets, in a process similar to the introduction of the chopstick electrodes in transwells (Figure 7c, left). This is a simple method and does not affect optical access to the cells inside the chip. However, it has low reliability due to variations in electrode placement and the small channel geometries [99]. Alternatively, electrodes can be integrated on-chip and placed closer to the cell culture chamber, reducing the resistance from the cell culture medium and the noise created by the electrode motion (Figure 7c, right) [86].

The successful integration of a tetrapolar electrode system into an OoC device is dependent on a careful choice of electrode size, geometry and placement. Odijk et al. reported problems resulting from the integration of electrodes on OoC, frequently resulting in an overestimation of TEER values [100]. Grimnes et al. showed that tetrapolar systems are vulnerable to errors [101]. In particular, the group showed that these systems typically have non-uniform sensitivity, with zones of positive, negative or zero sensitivity. If the sensitivity is negative, an increase in the electrical resistivity of a cell culture volume results in a decrease in the measured electrical resistance. Therefore, it is erroneous to assume that the entire cell culture area contributes equally to the TEER measurement. Finite element analysis can be performed to determine the current density and sensitivity field along the tissue barrier for specific electrode configurations and chamber geometries. Moreover, a geometric correction factor can be used for calculating TEER to account for non-uniform current densities.

Other potential sources of errors for TEER measurements performed on-chip include air bubbles present in microchannels and incomplete cell coverage. It has been reported that a slight gap in tissue coverage (0.4%) can result in the measured TEER values dropping by approximately 80% [100]. This problem can be particularly relevant for FTSm generated using animal-derived collagen, highly prone to shrinkage during culture. Therefore, accurate TEER measurements on a SoC device are highly dependent on the use of a mechanically stable dermal component.

## 4. The Evolution of Skin-on-a-Chip Platforms

OoC platforms for skin cultivation are still in their infancy, with the first paper in the field being published by Ataç et al. in 2013 [102]. Since then, multiple SoC platforms using a wide range of techniques have been reported. In this Review, we divide the existing platforms into transferred SoC, 2D SoC models, 3D SoC with perfusable lumens and 3D SoC with microfluidic channels (Figure 8). It is important to note that many of the included SoC models are far from the original definition of OoC due to their design and scale. In this Review, we expand the definition of SoC to include devices for producing skin tissues under dynamic perfusion, independently of the channel dimensions.

The most common application of SoC devices has been the maintenance of skin tissues under dynamic perfusion to increase their longevity and/or to establish a co-culture of tissues modeling different organs (Figure 8a). The skin tissues integrated on the chips are typically skin biopsies [103,104,105] or FTSm developed off-chip [102,106,107,108]. These studies provide valuable information regarding the use of SoC devices and their clinical purposes including multi-organ crosstalk and drug sensitivity and toxicity. However, the generation of the skin models outside the chip limits the benefits of the dynamic culture. This Review will focus on SoC devices for in situ generation of the tissues. We refer the reader to existing reviews for a detailed overview of the transferred SoC models [109].

### 4.1. Two-Dimensional Skin-on-a-Chip Models

One approach to mimic the human skin’s structure and functional responses consists of culturing layers of 2D cells on-chip, mimicking different skin compartments (Figure 8b, Table 1). Using this approach, Wufuer et al. proposed a model consisting of three layers (epidermal, dermal and vascular) co-cultured inside a SoC device [110]. A porous membrane separated each layer to allow interlayer communication. A model of skin inflammation was generated by perfusion with tumor necrosis factor (TNF-α), followed by measurement of proinflammatory cytokines levels and tight junction analysis. The efficacy of the drug dexamethasone was evaluated using the developed inflammation model. The study demonstrated that this drug could attenuate the effects of TNF-α including endothelial barrier dysfunction.

In the same year, Ramadan et al. described a SoC to develop an immune-competent in vitro skin model [111]. The model included a confluent layer of a human skin keratinocyte cell line (HaCaT), cultured on a porous membrane acting as a model of the epidermis and immune cells (leukemic monocyte lymphoma cell) positioned beneath the membrane. Silver/Silver chloride (Ag/AgCl) wires were inserted into the platform to measure the TEER, thereby monitoring the cell layer integrity through the course of cell culture and in response to chemical/physical stimuli. The group found that the perfusion-based culture promoted the barrier function and the lifespan of the cellular system. Furthermore, they investigated the effects of lipopolysaccharides (LPS) and the effects of UV radiation stimulus on the developed model. These experiments allowed the understanding of the role of the human keratinocyte layer as a protection barrier.

Sasaki et al. developed a photolithography-free device to culture an HaCaT monolayer and perform permeation assays [112]. In this work, the group developed a simple platform without using complex microfabrication techniques, which could be a barrier to some researchers. A porous membrane was sandwiched between branched microchannels and bonded using a PDMS mortar. The group tested the effect of potassium dichromate on the permeability of the cell monolayer by introducing fluorescein isothiocyanate-dextran (FITC-Dextran) solution on the top channel.

In the previously referred 2D SoC models, the skin cells were cultured directly in the microfluidic chip (in situ) to simulate the different skin components. These models have been successful in simulating diseases and their interaction with the immune system. However, they do not represent the complexity and 3D architecture of the native human skin.

### 4.2. SoC Models Based on 3D Cell Cultures

In the last years, various groups developed OoC devices for 3D skin tissue formation inside the platform. For this, two different approaches have been used; one based on the construction of dermal compartments with perfusable lumens, mimicking the vascular networks and the other based on the use of channels (Figure 8c), usually designed on a PDMS layer, below the skin tissue (Figure 8d).

#### 4.2.1. Models with Perfusable Lumens

FTSms with perfusable lumens have been generated using multiple 3D patterning techniques such as 3D printing, templating and sacrificial molding as well as cell-based approaches (Figure 8c, Table 2). The first reconstructed FTSm with a perfusable network was generated by Groeber et al. [113]. The group combined a biological vascular scaffold (decellularized segment of porcine jejunum) with a tailored bioreactor system to form a differentiated FTSm. Additionally, endothelial cells lined the walls of the development vessels to generate in vivo-like vasculature. However, the use of an animal organ limits the use of this model in large-scale production.

Abaci et al. developed a FTSm with vascular networks micropatterned by 3D printing sacrificial channels of cross-linked alginate [114]. These microchannels were embedded in the dermal compartment composed of collagen type I and were subsequently removed using sodium citrate after epidermal cornification. The group introduced either endothelial cells derived from iPSCs or HUVECs to cover the inner surface of these channels. These cells made it possible to recapitulate the endothelial barrier function, decreasing the permeability and diffusivity of the channels when compared to the control. Furthermore, the group grafted the vascularized human skin models to immunodeficient mice. It was possible to obtain neovascularization during wound healing, showing its potential for the treatment of cutaneous wounds.

Mori et al. fabricated a culture device using 3D templating techniques [115]. The group developed a device with anchoring structures and nylon wires strung across the connectors. A collagen structure was fixed into the device and perfusable vascular channels were created by removing the nylon wires. The group could recreate some features of the dermal/epidermal morphology and in vivo tight junctions on the vascular channel. However, this technique resulted in only one microchannel and lacked a microvascular network in the dermis. Moreover, the contraction of the collagen used for the dermal compartment affected the permeation assay, which had to be restricted to the central portion of the model.

An alternative approach for generating perfusable vascularized human skin models consists of the use of 3D bioprinting. Kim et al. used this technique for engineering a complex vascularized 3D human skin composed of epidermis, dermis and hypodermis [116]. For vasculature generation, a bioink composed of gelatine, glycerol, and thrombin with embedded endothelial cells was printed onto the hypodermal compartment; once finished, the construct was incubated at 37 °C, eliminating the gelatine and leaving hollow tubes inside the tissue. Proper tissue formation and maturation were reported, along with good vascular permeability properties. Still, the vasculature in the model was limited to one microchannel and lacked microvascularization.

Recently, Salameh et al. used 3D templating techniques to develop a vascularized FTSm that includes a more complex vasculature system than the previously described models [117]. The technique used to produce hollow channels inside the collagen matrix was similar to the work by Mori et al., using nylon wires [115]. However, to induce vasculogenesis, an additional step was included by seeding HUVECS transduced with Turbo-RFP lentiviruses (RFP-HUVECS). After removing the nylon wires, the hollow channels were seeded with HUVECS, and perfusion was initiated using a peristaltic pump. It was possible to generate a differentiated epidermis, perfusable vascular channels with angiogenic sprouts and an adjacent microvascular network. Furthermore, the potential of this model for topical and systemic applications was validated. Two compounds (caffeine and minoxidil) were topically applied to measure skin permeability and a pollutant (benzo[a]pyrene) was systemically applied.

#### 4.2.2. Models with Basal Perfusion

More recently, multiple SoC devices used microfluidic-based techniques to develop FTSms with basal perfusion (Figure 8d, Table 3). Lee et al. created a device with a simple structure and gravity-induced flow system to generate 3D FTSms [118]. The chip consisted of two layers of PDMS assembled on top of a glass base. The bottom PDMS layer consisted of the fluidic chamber, and the top PDMS layer had a central chamber for the skin model formation. A polycarbonate membrane was bonded between the bottom and top layers. The skin model was composed of collagen and primary human dermal fibroblasts mimicking the dermal compartment and primary human epidermal keratinocytes representing the epidermal layer. In addition, HUVECs were cultured on the bottom-side surface of the membrane to represent the vascular structure. The group observed comparable expression of keratin 5, involucrin and filaggrin when compared with conventional, transwell-based skin models. However, the *stratum corneum* of the perfused skin was less homogeneous than the conventional model. The authors hypothesized that this could be caused by an inhomogeneous distribution of nutrients and result in a “leaky” barrier function.

In the same year, Song et al. used the described platform to compare the shrinkage of collagen between static and dynamic systems [119]. The group also studied the expression of key marker proteins (fibronectin, collagen IV and keratin 10). They observed that the contraction of the hydrogel was lowest in the dynamic system. However, the markers were less expressed in the SoC model. A similar platform was also used for studying skin aging and the effect of different drugs and formulations. By inserting permanent magnets into a dedicated cavity on the PDMS layer and applying an external electromagnetic field Lim et al. could stretch the membrane attached to the polymer and develop a wrinkled skin [120].

Strüver et al. developed a perfusion platform to apply mechanical forces and shear stress to in vitro skin models [121]. They used bovine collagen and primary fibroblasts to develop the dermal compartment followed by the addition of primary keratinocytes on top of the matrix. After 1 day under submerged conditions, the skin models were transferred to the developed platform and grown for 6 days at the ALI. The device was constructed to be compatible with 6-well cell culture inserts. To assess the barrier function of the skin models, skin permeability studies were performed using reference substances (testosterone and caffeine). The perfusion resulted in the thickening of the *stratum corneum* and a higher compaction of the dermis equivalent. Furthermore, the group observed increased expression of filaggrin and involucrin following dynamic perfusion. However, it was not possible to observe improved barrier function on the dynamically cultivated skin by performing skin permeability. The group concluded that the perfusion of skin models might not be sufficient to produce an enhanced barrier function similar to in vivo skin.

Most of the previously mentioned studies included the use of animal-derived collagens to produce the dermal compartment. Consequently, they reported the poor mechanical properties resultant from the fibroblast-mediated matrix contraction and matrix degradation. These phenomena decreased reproducibility and limited the SoC lifespan as well as its applicability. Moreover, the groups also reported a lack of attachment of the culture to the membrane and chip walls due to acute shrinkage.

Recently, to overcome these problems, chemical and physical modifications of the matrix adding synthetic or natural polymers were proposed. Sriram et al. developed a model based on the use of fibrin that could overcome the typical limitations (e.g., low mechanical stability and contraction) of current SoC devices that use animal-derived hydrogel for the dermal compartment [122]. As both pure collagen and fibrin present poor mechanical stability, the group developed a modified biomaterial by combining fibrinogen with PEG. This mixture was combined with fibroblasts and the gelation was initiated by adding human thrombin. The volume of the final mixture to be pipetted into the SoC device was optimized to produce a flat surface for keratinocyte seeding. The device was composed of a multi-chamber microfluidic chip consisting of two fluidic compartments separated by a permeable microporous membrane. The experimental setup also included interchangeable lids and insets to switch from a bioreactor to an in vitro analysis system. Using the developed device, the group generated a stratified epidermis with an enhanced basement membrane. In particular, the deposition of collagen IV, VII and XVII were enhanced when compared with the control. The TEER values were also higher than the controls, pointing to an enhanced barrier function.

Recently, a different approach for the development of a stable and physiologically relevant dermal compartment was presented by Zoio et al. [97,124]. The group used rapid prototyping techniques to develop a modular device with integrated electrodes for TEER measurement. The reported approach combined the production of a fibroblast-derived matrix (FDM) with an inert polystyrene porous scaffold integrated on-chip, excluding exogenous hydrogels and membranes. TEER was measured in situ during skin culture and to evaluate the impact of a benchmark irritant onto the skin barrier. The dynamic flow resulted in increased synthesis and deposition of FDM proteins (collagen I and fibronectin). The developed SoC presented increased thickness and enhanced barrier function compared to the controls.

Valencia et al. developed a 3D skin model including a dermal and epidermal compartment using an innovative approach [123]. The group developed a controlled parallel flow method to generate a bilayer tissue by using syringe pumps. The HaCaT cell line and HFs were used to generate the model. In a similar approach to Sriram et al., the group used human fibrinogen to form a fibrin hydrogel for the dermal compartment. Thrombin and tranexamic acid were added to the fibrinogen to prepare a pre-gel (non-gelled fibrinogen solution). The device was fabricated using an edge plotter and included two PMMA layers, one PDMS layer, vinyl upper and lower chambers and an integrated polycarbonate membrane. Contrary to conventional systems where cells are manually pipetted into the culture channels, the group was able to inject all components (cells and ECM) directly into the channels using syringe pumps. With the developed approach, it was possible to obtain a 3D dermal compartment with HFs and HKs on top to simulate the epithelial compartment. Although this is a promising approach that could increase automatization and standardization, no differentiated epidermis was produced, and no air-liquid interface was established.

Finally, Rimal et al. developed a 3D vascularized FTSm with basal dynamic flow using a custom-built 3D-printed bioreactor [125]. The group generated a scaffold-free dermal compartment using cell coating techniques. Primary fibroblasts were coated with a thin layer of ECM (fibronectin and gelatin) and mixed with endothelial cells. Primary keratinocytes were added on top of the generated dermal compartment. The group reported enhanced epidermal barrier properties and ECM deposition (fibronectin and collagen I) under dynamic flow. Furthermore, flow culture resulted in a significant increase in skin thickness compared to static cultures. The group tested the applicability of the developed model by performing 3D wound healing assays in the absence and presence of flow. A faster wound healing in flow culture was observed.

## 5. Comparative Analysis of SoC Devices

Taken together, these studies demonstrate the potential of SoC platforms for a multitude of applications ranging from disease modeling to drug testing. However, by comparing the reports from the different groups, inconsistencies regarding the effects of the OoC technology on the skin architecture and physiology can be observed (Table 4). Some groups report negative effects from developing skin models inside the chip, with dynamic perfusion. This is the case of the work from Lee et al. in which dynamic perfusion resulted in a more uneven *stratum corneum* than the static model [91]. Moreover, Song et al. observed that collagen IV, a primary collagen found in the basement membrane, was less expressed in the SoC model [119]. Other groups reported a comparable architecture between SoC and conventional skin models. Mori et al. developed a perfused FTSm with a comparable thickness and differentiation profile to the controls [115].

Several recent works pointed out the benefits of dynamically cultured models. The most commonly reported benefit of OoC technology on epidermal markers is the increased expression of filaggrin and involucrin, proteins with important roles in the formation of the epidermal skin barrier [121,122,124,125]. Additionally, Sriram et al. reported clear benefits of the dynamic culture on the synthesis of basement membrane proteins (collagen IV, VII and XVII) [122]. None of these studies reported a difference between SoCs and static models concerning keratin 10. The most commonly reported benefit of the dynamic perfusion on dermal markers is the increased expression of fibronectin [124,125]. No consensus can be observed concerning the expression of collagen. Finally, functional studies point to OoC technology enhancing the skin’s barrier function. The most common finding was an increased SoC mean TEER compared to the static controls [122,124,125]. The higher TEER seen in the SoC models translates to a lower permeability of topically-applied compounds (caffeine [122] and FITC-dextran [124]). An opposite finding was reported by Strüver et al. [121]. The group showed increased permeability for a lipophilic test compound (testosterone) in the dynamically cultured skin models. They hypothesized that these findings could result from collagen detachment and concluded that simple perfusion of skin models might not be sufficient to enhance barrier function.

The different results obtained by the groups developing SoC devices can be explained by their widely different approaches regarding cell type, dermis composition, scaffold material and shape, ECM matrix, chip design and type of flow. In particular, the nature of cells used in the FTSm can compromise the ability to generate fully differentiated SoC models and their overall reproducibility. Several researchers use primary cells to generate their SoC models, which is considered a key point for the development of physiologically relevant in vitro skin. However, these cells present limited population doublings and limited reproducibility. Other groups report the use of cell lines, such as HaCaT cells, without successful reproduction of a 3D differentiated structure. At present, Sriram et al. is the only group that developed a SoC using TERT-immortalized keratinocytes. It would be important to understand, for example, if the effects of perfusion and mechanical forces on the generated SoC depend on the types of cells used.

An important parameter that also extensively varies in the reported studies and that can affect the final structure and applicability of the SoC model is the material used for generating the dermal compartment. The in vivo ECM comprises multiple collagens, various fibrous proteins and proteoglycans. Most studies report SoC models with dermal compartments comprising exclusively collagen type I. This is not representative of the in vivo-like microenvironment and normally results in fibroblast-mediated matrix contraction and matrix degradation. This phenomenon limits their lifespan and reproducibility. Furthermore, collagen detachment inside the SoC platform can limit the maintenance of a leak-free fluid-tissue-air barrier. Therefore, the use of animal-derived collagen to generate several SoC models can explain part of the inconsistencies.

Other relevant parameters should be further investigated and fine-tuned towards developing reproducible and standardized SoC models. This is the case of the perfusion rates, which range from 0.6 μL/min to 5 × 10^3^ µL/min, depending on the proposed experimental setup. Air flow and gas composition of the skin compartment exposed to the air can also be relevant parameters to produce models with improved stratification and homeostasis. This differs between the studies with some groups employing active ventilation using peristaltic pumping and others employing passive ventilation.

## 6. Conclusions and Future Perspectives

Although the use of SoC devices to generate biomimetic skin is still in its infancy, much progress has been made in this field. These devices have been used to simulate diseases, bacterial infections and to test drugs regarding toxicity and efficacy. Furthermore, the first steps have been taken to establish a co-culture of different tissues to study the systemic effect of topically applied drugs. However, further improvements are needed for the successful translation of the SoC models to industry and clinical settings.

First, for these SoC devices to become mainstream tools in biology laboratories, it is important to solve current design, manufacturability and usability challenges. Most of the mentioned works used traditional lithography technology to prototype PDMS devices, limiting their potential to be mass-produced. Alternative micropatterning techniques and materials such as PMMA, PVC or polycarbonate should be considered. Future standardization of chip design, materials and manufacturing techniques would also greatly simplify cross-lab validation of these devices. Furthermore, standardization of OoCs would facilitate their integration within existing technology and equipment (e.g., conventional microscopes, incubators). Compatibility with laboratory equipment will be critical for the adoption of SoC models by the scientific community. Importantly, these platforms should be self-sufficient, robust and simple to operate, allowing multiple specimens to be cultured in parallel and reducing the production costs. Sensor integration should also be considered depending on the particular context and research goal. For drug testing applications, SoCs with integrated sensors could allow real-time monitoring of multiple parameters, in real-time, following the administration of drugs. In particular, TEER measurements could be a promising parameter to monitor skin barrier function, for quality control and to evaluate the effect of test drugs.

Regarding the biological components, the dermal and epidermal compartments should reflect the biophysical properties of the native microenvironment. Furthermore, the dermal compartment should remain stable during cell culture to avoid shrinkage and detachment to the culture insert. This can be achieved through chemical or physical modifications of natural polymers. In particular, the combination of fibrinogen and PEG can be a promising approach to generate stable and physiologically relevant dermis structures for a successful model. Alternatively, dermal cells can be stimulated to produce endogenous ECM, excluding the use of animal-derived materials. This strategy could be combined with a scaffold structure to give mechanical stability during tissue culture inside the chip. Importantly, reproducible cell and ECM matrix loading should be achieved to generate a robust SoC device. Reproducible loading along with the prevention of trapped air bubbles inside the device are considered the most relevant features by end-users of OoC, therefore important to achieve broad adoption [126]. The most common approach to load cellular components into the SoC is by manually pipetting the cells into the device. Increased robustness could be achieved by automatization and standardization of the cell/ECM matrix loading process, avoiding manual seeding procedures.

The cell type and origin are also relevant parameters for the development of fully differentiated FTSms. Primary cells have demonstrated the ability to produce FTSms with in vivo-like architecture. Immortalized human N/TERT keratinocytes closely resemble primary keratinocytes in their ability to stratify. The use of immortalized cells could be considered to increase the reproducibility of the model. However, more studies are required to assess the physiological relevance of these cells. Furthermore, current biomimicry is limited by the cells and appendages incorporated in the SoC. Additional cell types such as melanocytes, Merkel cells and immune cells as well as appendages such as sweat glands could also be integrated to expand the applicability of future models. However, the trade-off between complexity and reproducibility needs to be considered.

Additionally, iPSCs derived from specific patients could become an important tool for personalized medicine to model healthy and disease characteristics tailored to the patient. Considering the high costs and low yield and reproducibility when using iPSC, lab automation should be implemented in parallel. Finally, the medium used to perfuse the tissue should be serum-free to reduce the use of animal-derived materials and associated batch variability issues.

The development of physiologically relevant SoC models will be crucial to obtain a useful tool to study the efficacy and toxicity of skin-targeted drugs and to evaluate the systemic effects of pharmaceutical compounds if connected with other OoCs (multi-OoC). Moreover, SoCs are promising tools to model both physiological and pathological skin conditions, including skin cancer, expanding our comprehension of skin biology and contributing to the development of better drugs.

## Figures and Tables

**Figure 1 pharmaceutics-14-00682-f001:**
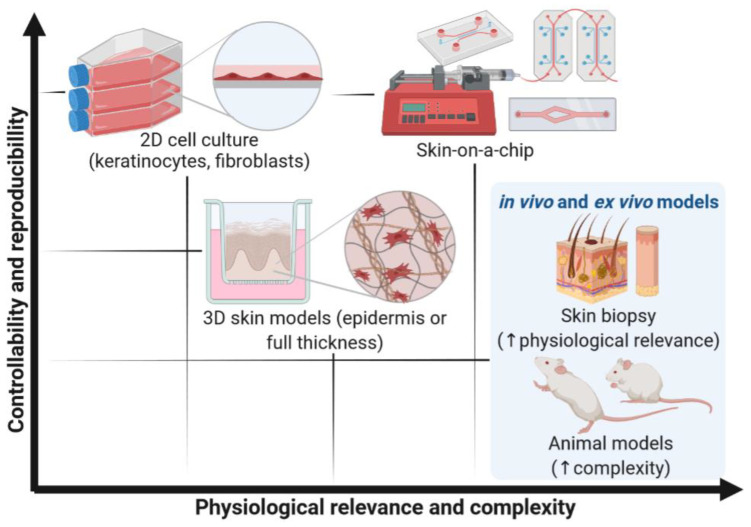
Methods used for preclinical studies of skin-targeted drugs. Comparison between the different methodologies regarding controllability and reproducibility as well as physiological relevance and complexity. The figure highlights the potential significance of organ-on-a-chip (OoC) technology to provide a reproducible and physiologically relevant approach to systematically evaluating skin responses.

**Figure 2 pharmaceutics-14-00682-f002:**
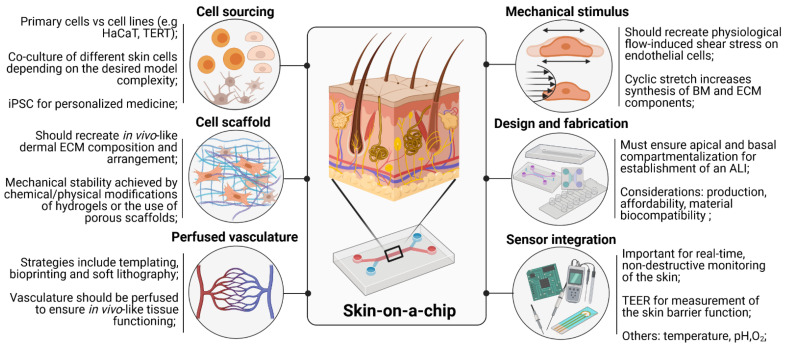
Development of biomimetic skin-on-a-chip platforms. Schematic drawing representing the main factors to be considered when developing physiologically relevant skin-on-a-chip (SoC) models, including technical and biological factors (cell sourcing, cell scaffold, perfusion, cyclic stretching, design and fabrication and sensor integration). The different factors should be evaluated taking into account the specific application as well as available equipment and know-how. ALI: Air-liquid interface; BM: Basement membrane; ECM: Extracellular matrix; TEER: Transepithelial electrical resistance; iPSC: Induced pluripotent stem cells.

**Figure 3 pharmaceutics-14-00682-f003:**
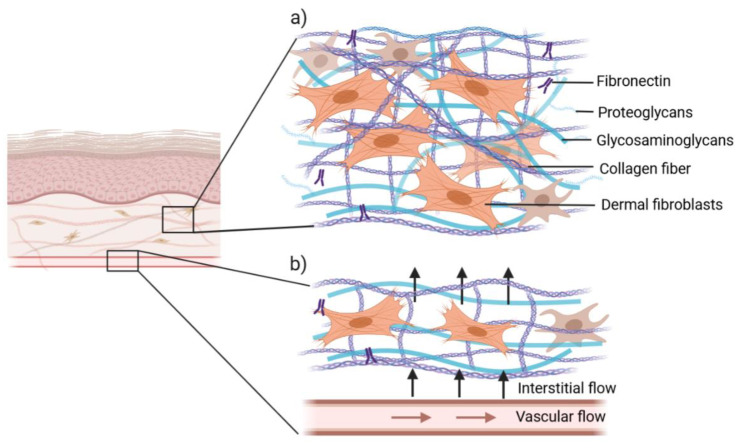
Extracellular matrix (ECM) in vivo skin. (**a**) Schematic of the dermal matrix showing dermal fibroblasts surrounded by the components of typical ECM, composed of collagens, glycosaminoglycans, proteoglycans and fibronectin. These components are secreted and remodeled by the fibroblasts, which results in an organized meshwork. (**b**) Schematic of the interstitial flow. The interstitial flow provides convection necessary for transport of molecules through the ECM or *interstitium* and induces morphogenic effects in dermal cells.

**Figure 4 pharmaceutics-14-00682-f004:**
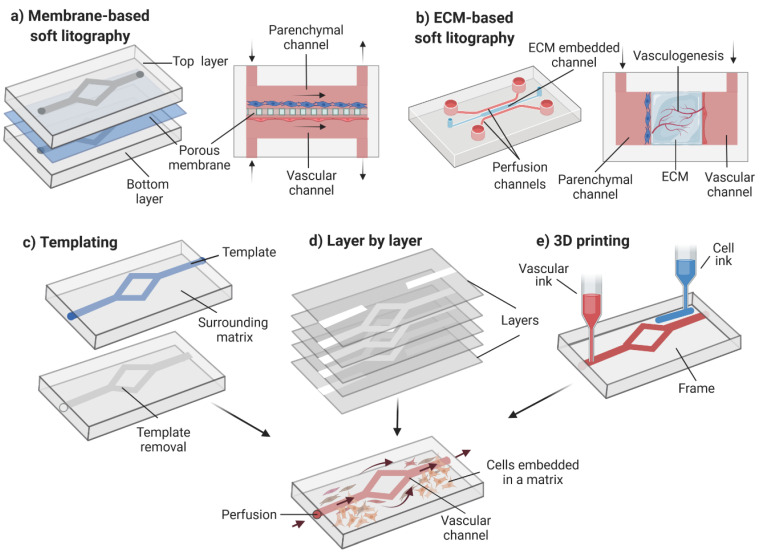
Schematic representation of techniques used to vascularize 3D OoC models. These techniques can be divided into soft-lithography and 3D patterning approaches. (**a**) Membrane-based soft lithography technique in which a porous membrane is sandwiched between two microfluidic layers. (**b**) ECM-based soft lithography in which one or more channels are filled with ECM, embedded between the parenchymal and vascular channels. (**c**) Templating approaches in which a matrix is cast around the template. The template is subsequently removed, generating hollow channels, which can be seeded and perfused. (**d**) Layer-by-layer approach consisting of assembled modular layers. (**e**) Three-dimensional printing (bioprinting) in which vascular and cell inks are used to generate a 3D tissue with embedded, perfusable vascular channels.

**Figure 5 pharmaceutics-14-00682-f005:**
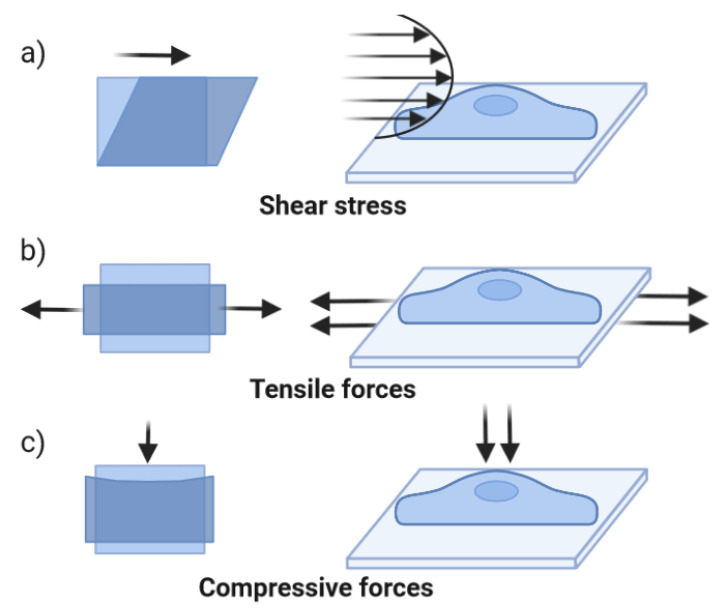
Different mechanical stress types acting on cells, including (**a**) shear stress, (**b**) tensile (stretch) forces and (**c**) compressive forces Simplified illustrations of the effect of each force in planar culture shown for an idealized square (**left**) and a cell (**right**).

**Figure 6 pharmaceutics-14-00682-f006:**
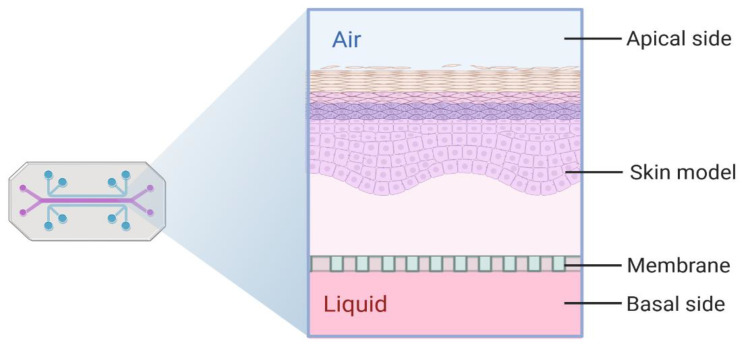
Schematic representation of a polymeric OoC with integrated porous membrane for establishing two separated compartments (apical and basal). A FTSm is generated on top of the membrane and maintained at the ALI with basal perfusion.

**Figure 7 pharmaceutics-14-00682-f007:**
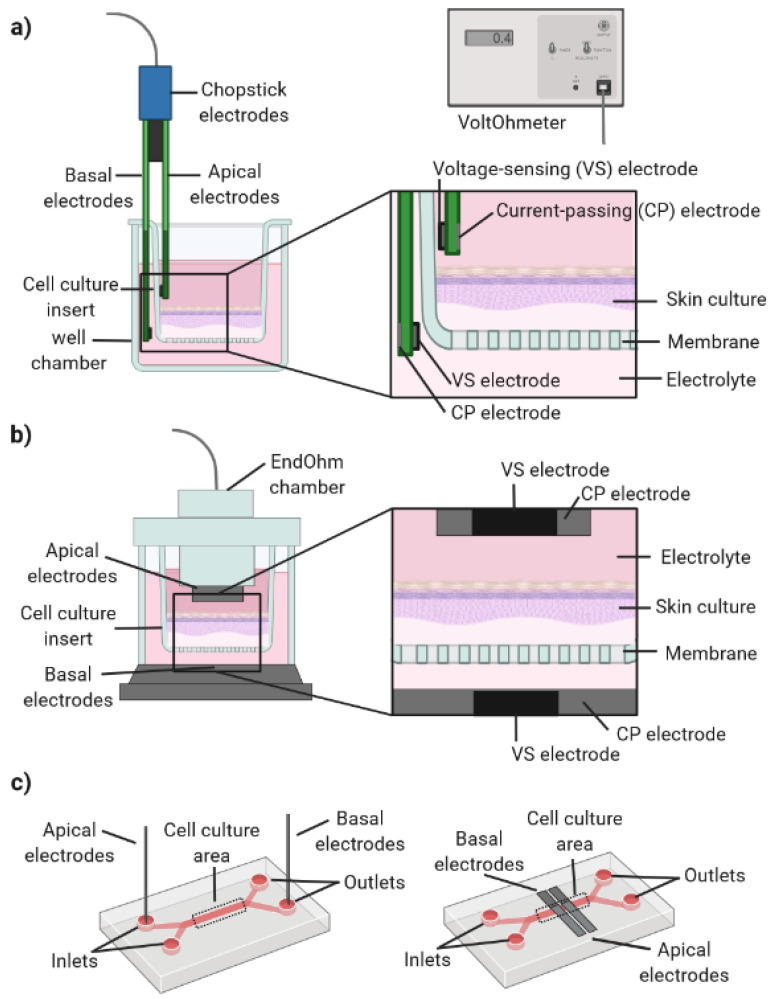
Transepithelial/transendothelial electrical resistance (TEER) measurements. (**a**) Using chopstick/STX2 electrode pairs connected to voltOhmeter and (**b**) using an EndOhm chamber. The uniformity of the current density generated by the electrodes across the cell layer has a significant effect on the TEER measurement. The chopstick electrodes cannot deliver uniform current density. As an alternative, the EndOhm chamber generates a more uniform current density. (**c**) TEER measurements performed on-chip by inserting electrodes on the chip inlets/outlets (**left**), in a process similar to the introduction of the chopstick electrodes. Electrodes integrated on-chip and placed closer to the cell culture chamber (**right**), in a process similar to the EndOhm chamber.

**Figure 8 pharmaceutics-14-00682-f008:**
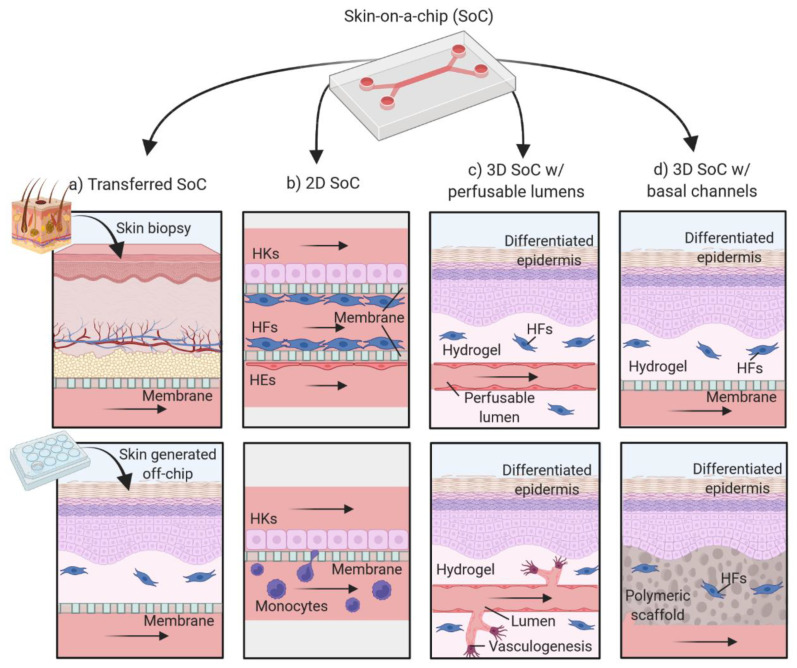
Schematic diagram of the strategies used to develop SoC models. (**a**) Transferred SoC devices in which skin biopsies (**top**) or skin models generated off-chip (**bottom**) are transferred on-chip. (**b**) 2D SoC devices in which cell monolayers are cultured on porous membranes, establishing different compartments. Two-dimensional SoC with 2 membranes (3 compartments) (**top**) and 1 membrane (2 compartments) (**bottom**) (**c**) 3D SoC with perfusable lumens (**top**). The perfusable lumens are created using 3D patterning techniques, typically using templating, sacrificial modeling or bioprinting. This approach can be combined with vasculogenesis (**bottom**). (**d**) 3D SoC with microfluidic channels. Membrane-based SoC device in which the microfluidic layers and a porous membrane are assembled in a sandwiched stricture (**top**). The skin model is cultured in situ on top of the membrane. Alternatively, a porous scaffold can be integrated, excluding the use of the membrane and hydrogel. Abbreviations: HKs, human keratinocytes; HFs, human fibroblasts, HEs, human endothelial cells.

**Table 1 pharmaceutics-14-00682-t001:** Summary of the 2D SoC devices reported in the literature. Abbreviations: HUVECS, human umbilical vein endothelial cells; PDMS, polydimethylsiloxane; PEG, polyethylene glycol; PET, polyethylene terephthalate; PMMA, poly(methylmethacrylate); TEER, transepithelial electrical resistance; TNF, tumor necrosis factor.

Reference	Cell Type	Flow Type; Velocity	Fabrication Method; Materials	Main Features
Wufuer et al. (2016) [110]	HaCaT, immortalized HS27 and HUVECS	Gravity driven flow; Not stated	Photolithography; PDMS and PET membrane	TNF-α induced skin inflammation; Simulation of skin edema
Ramadam et al. (2016) [111]	HaCaT and U937 dendritic cells	Syringe pumping; 0.6–1.2 µL/min	Rapid prototyping; PMMA, PDMS and PET membrane	Immune competent model; TEER measurements
Sasaki et al. (2019) [112]	HaCaT	Syringe pumping; 10 µL/min	Laser cutter; PMMA, PDMS and PET membrane	Irritation testing with potassium dichromate

**Table 2 pharmaceutics-14-00682-t002:** Summary of the 3D SoC with perfusable lumens reported in the literature. Abbreviations: HEKs, human epidermal keratinocytes; HDFns, human dermal fibroblasts; iPSC, induced pluripotent stem cells; hDMECs, human dermal microvascular endothelial cells; HUVECS, human umbilical vein endothelial cells; HPAs, human preadipocytes subcutaneous; dECM, decellularized extracellular matrix; PEEK, polyether ether ketone; PC, polycarbonate; PET, polyethylene terephthalate; PCL, polycaprolactone; TEER, transepithelial electrical resistance; ISDN, isosorbide dinitrate.

Reference	Cell Type	Dermal Matrix	Flow Type; Velocity	Fabrication Method; Materials	Main Features
Groeber et al. (2016) [113]	Primary HEKs, primary HDFs and hDMECC	Decellularized porcine jejunum	Peristaltic pumping; Not stated	Rapid prototyping; PEEK, PC	TEER measurements,
Abaci et al. (2016) [114]	Primary HEKs, primary HDFs iPSC-derived endothelial cells	Collagen	Syringe pumping; Not stated	3D printing, templating; Tranwell insets, PET membranes	Integration of iPSC; Promotion of neovascularization in a rat model
Mori et al. (2016) [115]	Primary HEKs, primary HDFs and HUVECs	Collagen	Peristaltic pumping; 33–50 µL/min	3D printing, templating; Not stated	Vascular channels coated with endothelial cells; Permation testing caffeine and ISDN
Kim et al. (2019) [116]	Primary HEKs, primary HDFs, HUVECs and primary HPAs	Fibrinogen, dECM porcine skin	Peristaltic pumping; 50–100 µL/min	3D printing; PCL	Bioprinting; Integration of hypodermis
Salameh (2021) [117]	Primary HEKs, primary HDFs and HUVECs	Collagen	Peristaltic pumping; 33–50 µL/min	3D printing, templating; Not stated	Formation of angiogenic sprouts; systemic drug delivery studies

**Table 3 pharmaceutics-14-00682-t003:** Summary of the 3D SoC with basal perfusion reported in the literature. Abbreviations: HEKs, human epidermal keratinocytes; HDFns, human dermal fibroblasts; HUVECS, human umbilical vein endothelial cells; PEG, polyethylene glycol; FDM, fibroblast-derived matrix; PS, polystyrene; ECM, extracellular matrix; FN, fibronectin; G, gelatin; PDMS, polydimethylsiloxane; PC, polycarbonate; PTFE, polytetrafluoroethylene; PET, polyethylene terephthalate; PMMA, poly(methylmethacrylate); FITC, fluorescein isothiocyanate; TEER, transepithelial electrical resistance.

Reference	Cell Type	Dermal Matrix	Flow Type; Velocity	Fabrication Method; Materials	Main Features
Lee et al. (2017) [118]	HaCaT or primary HEKs, primary HDFs and HUVECs	Collagen	Gravity driven; 5–10 µL/min	Lithography; PDMS and PC membrane	Studies of mass transport with FITC-dextran
Song et al. (2017) [119]	Primary HEKs and primary HDFs	Collagen	Gravity driven; Not stated	Lithography; PDMS and PC membrane	Study of collagen contraction
Lim et al. (2018) [120]	Primary HEKs and primary HDFs	Collagen	Gravity driven; Not stated	Lithography; PDMS and PC membrane	Uniaxial stretch applied for modeling wrinkles
Strüver et al. (2017) [121]	Primary HEKs and primary HDFs	Collagen	Peristaltic pumping; 20–167 µL/min	Not stated; PTFE and PET membrane	Improved skin differentiation
Sriram et al. (2019) [122]	N/TERT and primary HDFs	Fibrin + PEG	Peristaltic pumping; 1 µL/min	Micromilling; PMMA and PC membrane	Improved skin differentiation and barrier function; stable dermis
Valencia et al. (2021) [123]	HaCaT and primary HDFs	Fibrin	Syringe pumping; 0.67 µL/min	Edge plotter; PMMA, PDMS, vinyl and PC membrane	Parallel flow method for bilayer tissue formation
Zoio et al. (2021, 2022) [97,124]	Primary HEKs and primary HDFs	FDM + PS scaffold	Syringe pumping; 1–2 µL/min	Rapid prototyping; PMMA	Improved barrier function; TEER measurements on-chip
Rimal et al. (2021) [125]	Primary HEKs, primary HDFs and HUVECS	ECM-coating of single cells (FN and G)	Peristatic pumping; 5 × 10^3^ µL/min	3D printing; Not stated	Scaffold-free; vascularized dermal tissue; 3D-wound healing assay

**Table 4 pharmaceutics-14-00682-t004:** Comparison between SoC models and static controls reported in the literature. ↑ represents increased physiological relevance of the SoC model compared to the controls, ↓ represents decreased physiological relevance of the SoC model compared to the controls, and =represents no significant difference between the SoC model and the controls. Abbreviations: DEJ, dermo-epidermal junction; K, keratin; VE, viable epidermis; SC, stratum corneum; TEER, transepithelial resistance; FD, FITC-Dextran; ECM, extracellular matrix.

Reference	Epidermis and DEJ Markers	Dermis Markers	Thickness	Functional Studies	Others
Mori et al. (2016) [115]	=K10, =K15	-	=epidermis (VE + SC)	=capacitance	↑ cross-sectional area channels, ↑ cell density
Kim et al. (2019) [116]	↑ K19	-	-	-	↑ p63-positive cells
Lee et al. (2017) [118]	=K5, =involucrin, =filaggrin	-	-	-	↓ SC homogeneity
Song et al. (2017) [119]	↓ collagen IV, =K10	=fibronectin, ↓collagen	-	-	↓ hydrogel contraction
Strüver et al. (2017) [121]	↑ filaggrin, ↑ involucrin	-	↑ SC, =VE, ↓ dermis	↓ barrier function (increased testosterone permeability)	↑ claudin 1, ↑ occludin
Sriram et al. (2019) [122]	↑ collagen IV, ↑ involucrin, ↑ collagen VII, ↑collagen XVII, =filaggrin, =K10	-	↑ SC, ↑ VE	↑ TEER ↑ Barrier function (decreased caffeine permeability)	↓ SC water content
Zoio et al. (2022) [124]	=K10, =K14, ↑ filaggrin, ↑ involucrin	↑ collagen I, ↑ fibronectin	=SC, ↑ VE	↑ TEER ↑ barrier function (decreased FD permeability)	-
Rimal et al. (2021) [125]	↑ filaggrin, =filaggrin 2	↑ fibronectin, =collagen I	-	↑ TEER	↑ wound healing ↑ ECM homeostasis

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
