# Peer review of "Skin-on-a-Chip Technology: Microengineering Physiologically Relevant In Vitro Skin Models"

_pharmaceutics, 2022, doi:10.3390/pharmaceutics14030682_

Round 1
Reviewer 1 Report
The authors have provided a relatively comprehensive review of skin-on-a-chip technology. Although there are many recent review papers published in this field (the authors referred to most of them in their manuscript), the present review can complement these works and give interested researchers good insights into the design and fabrication of such platforms for in vitro studies.
To further improve the quality of the manuscript, I have the following suggestions:
1) It would be great if the authors assign a separate section explaining the possible challenges for commercialising such devices and briefly provide solutions to overcome these difficulties in real clinical settings.
2) The authors can briefly explain the possibility of integrating skin-on-chip models with microneedles for drug delivery and ISF sensing applications. Please see the following papers: https://doi.org/10.1016/j.apsb.2019.03.007 and https://doi.org/10.3390/chemosensors9040083
3) A brief discussion regarding the mathematical modellings and CFD simulations in skin-on-a-chip platforms (such as diffusion and porous media models) would be helpful, especially for the engineering audience. The authors can refer to this paper for details: https://doi.org/10.3390/mi12030294
Author Response
First, we want thank the reviewer for its time and insightful feedback. Bellow, we provide the point-by-point responses. The line numbers were counted without track changes activated:
- It would be great if the authors assign a separate section explaining the possible challenges for commercialising such devices and briefly provide solutions to overcome these difficulties in real clinical settings.
Author response: We agree with the reviewer on the importance of considering the current challenges for commercialization of SoC devices. In section 7, conclusions and future perspectives, we discuss the current challenges that need to be overcome for the clinical translation of SoC devices and provide some possible solutions. We added more information to the section to provide a more complete analysis of the translation problem:
(Lines 973 – 980)“...However, further improvements are needed for the successful translation of the SoC models to industry and clinical settings. First, for these SoC devices to become mainstream tools in biology laboratories, it is important to solve current design, manufacturability and usability challenges.”
(Lines 983 – 990) “Future standardization of chip design, materials and manufacturing techniques would also greatly simplify cross-lab validation of these devices. Furthermore, standardization of OoCs would facilitate their integration within existing technology and equipment (e.g., conventional microscopes, incubators). Compatibility with the lab’s equipment will be critical for the adoption of SoC models by the scientific community. Importantly, these platforms should be self-sufficient, robust and simple to operate; allowing multiple specimens to be cultured in parallel and reducing the production costs.”
(Lines 1006 – 1011) “Importantly, reproducible cell and ECM matrix loading should be achieved to generate a robust SoC device. Reproducible loading along with the prevention of trapped air bubbles inside the device are considered the most relevant features by end-users of OoC, therefore important to achieve broad adoption [127]. The most common approach to load cellular components into the SoC is by manually pipetting the cells into the device. Increased robustness could be achieved by automatization and standardization of the cell/ECM matrix loading process, avoiding manual seeding procedures. “
- The authors can briefly explain the possibility of integrating skin-on-chip models with microneedles for drug delivery and ISF sensing applications. Please see the following papers: https://doi.org/10.1016/j.apsb.2019.03.007 and https://doi.org/10.3390/chemosensors9040083
Author response: We thank the reviewer for pointing out this interesting application for skin-on-a-chip devices. As suggested by the reviewer, we briefly explained the possibility of integrating microneedles. Furthermore, we reference the papers suggested by the reviewer.
(Lines 508 – 513) “Furthermore, different modules could be developed, enabling the user to select the modules that meet their requirements. For example, a dedicated apical module with integrated microneedles could be developed to study the effect of these devices on the delivery of pharmacologically active ingredients. Microneedle devices are increasing in popularity due to their potential for transdermal delivery, disease treatment and diagnosis [78,79].”
- Yang, J.; Liu, X.; Fu, Y.; Song, Y. Recent advances of microneedles for biomedical applications: drug delivery and beyond. Acta Pharm. Sin. B 2019, 9, 469–483, doi:https://doi.org/10.1016/j.apsb.2019.03.007.
- Kashaninejad, N.; Munaz, A.; Moghadas, H.; Yadav, S.; Umer, M.; Nguyen, N.-T. Microneedle Arrays for Sampling and Sensing Skin Interstitial Fluid. Chemosens. 2021, 9.
3. A brief discussion regarding the mathematical modellings and CFD simulations in skin-on-a-chip platforms (such as diffusion and porous media models) would be helpful, especially for the engineering audience. The authors can refer to this paper for details: https://doi.org/10.3390/mi12030294
Author response: We agree with the reviewer that it could be useful to mention the importance of mathematical modelling and CFD simulations to optimize the design of the skin-on-a-chip platformed. We added some paragraphs discussing this topic. Furthermore, as suggested, we refer the reader to the paper shared by the reviewer.
(Lines 516 – 543) “Experimental studies can be complemented with numerical simulations to assess the viability of the SoC devices and optimize their design. These tools can be used to predict critical parameters such as oxygen concentration, fluid velocity, shear stress and diffusion processes. Consequently, it is possible to better determine the required chip materials, chamber/channel geometry and dimensions, fluid properties and testing conditions to develop physiologically relevant models.
Hernando et al showed the importance of performing in silico studies for design optimization by modelling cell behaviours in OoC devices with different channel geometries. The group reported that the cell culture time required to fully exploit their OoC could be reduced by redesigning the chip’s inlet channel and chamber network [80]. Zahorodny-Burke et al used numerical simulations to evaluate the impact of the chip’s materials on the oxygen concentration in cell culture. The group reported that OoC devices fabricated using PMMA and cyclic olefin copolymer (COC) resulted in a low oxygen supply to the cells. They concluded that flow rates should be optimized to increase oxygen supply when using materials with low oxygen diffusion [81]. More recently, Kheiri et al used computational modelling to simulate multiple device designs and flow conditions for reproducing tumor spheroid-on-a-chip [82]. The group concluded that computational modelling is an efficient strategy to optimize microfluidic device designs, providing insightful information on the drug transport phenomena in these models.
Similarly, numerical simulations can be used to simulate the microenvironments of SoC devices, towards developing models of greater accuracy. The adequate simulation of flow and drug transport in these models requires an understanding of the key physical properties of the skin. Narasimhan et al described a detailed numerical model for transdermal drug delivery by considering the skin as a composite, porous material [83]. The authors provided the equations governing the diffusion through the porous medium in the different skin layer. A detailed description of mathematical models that can be used to predict fluid movement and drug diffusion across the SoC models is also provided in a recent publication by Ponmozhi et al [84].”
- Ballesteros Hernando, J.; Ramos Gómez, M.; Díaz Lantada, A. Modeling Living Cells Within Microfluidic Systems Using Cellular Automata Models. Sci. Rep. 2019, 9, 14886, doi:10.1038/s41598-019-51494-1.
- Zahorodny-Burke, M.; Nearingburg, B.; Elias, A.L. Finite element analysis of oxygen transport in microfluidic cell culture devices with varying channel architectures, perfusion rates, and materials. Chem. Eng. Sci. 2011, 66, 6244–6253, doi:https://doi.org/10.1016/j.ces.2011.09.007.
- Kheiri, S.; Kumacheva, E.; Young, E.W.K. Computational Modelling and Big Data Analysis of Flow and Drug Transport in Microfluidic Systems: A Spheroid-on-a-Chip Study . Front. Bioeng. Biotechnol. 2021, 9.
- Narasimhan, A.; Joseph, A. Porous Medium Modeling of Combined Effects of Cell Migration and Anisotropicity of Stratum Corneum on Transdermal Drug Delivery. J. Heat Transfer 2015, 137, doi:10.1115/1.4030923.
- Ponmozhi, J.; Dhinakaran, S.; Varga‐medveczky, Z.; Fónagy, K.; Bors, L.A.; Iván, K.; Erdő, F. Development of skin‐on‐a‐chip platforms for different utilizations: Factors to be considered. Micromachines 2021, 12, 1–25, doi:10.3390/mi12030294.
Reviewer 2 Report
Physiologically relevant in vitro human skin models is highly needed for basic and translation studies. In this manuscript, Zoio and Oliva comparatively summarized the recent progresses of skin-on-a-chip (SoC) devices and discussed the technical and biological challenges when engineering a SoC device with physiologically relevant microenvironment of skin. The manuscript is comprehensive and well-written. Moreover, Figures are pretty informative. The Reviewer strongly recommends its publication in Pharmaceutics.
Below are some minor issues that need to be addressed.
- Figure 1 looks similar to the Figure 1 from Chen et al., Trends in Pharmacological Sciences, 2020. Reference needs to be cited.
- There are some typo. For example, Line 641, “in-cludeair bubbles present in microchannels and incomplete cell coverage”; Line 794, “However, the stratum corneum of the perfused skin was less homogeneous that the one from the conventional model.”
Author Response
First, we want thank the reviewer for recommending the publication of our manuscript. We have carefully considered the comments and tried our best to address every one of them.
Bellow, we provide the point-by-point responses. The line numbers were counted without track changes activated:
- Figure 1 looks similar to the Figure 1 from Chen et al., Trends in Pharmacological Sciences, 2020. Reference needs to be cited.
Author response: We added the suggested reference to the introductory text discussing the importance of OoC technology:
(Lines 92 -95) “The need for physiologically relevant and functional tissue models led to new technologies for cell cultures such as organ-on-a-chip (OoC) or microphysiological systems. This modern technology aims to surpass the limitations of the 3D cell-based culture platforms and increase the predictive power of in vitro models (Figure 1) [13].”
- Ma, C.; Peng, Y.; Li, H.; Chen, W. Organ-on-a-Chip: A New Paradigm for Drug Development. Trends Pharmacol. Sci. 2021, 42, 119–133, doi:10.1016/j.tips.2020.11.009
However, our figure 1 has key differences from the figure in the mentioned publication. Their illustration is more general and ours focuses on skin models. Furthermore, we don’t fully agree on their relative positioning of 3D and 2D cell culture on the graph. From their illustration, it seems that 3D models have the same physiologically relevance of 2D models, which is not the case. We also do not fully agree that organ-on-a-chip models currently have the same complexity as animal models. Still, we believe this is a very valuable publication and included the reference in our text.
- There are some typo. For example, Line 641, “in-cludeair bubbles present in microchannels and incomplete cell coverage”; Line 794, “However, the stratum corneum of the perfused skin was less homogeneous that the one from the conventional model.”
Author response: We want to thank the reviewer for pointing out these typos. We’ve corrected them.
Reviewer 3 Report
This review is an interesting, timely, well written and comprehensive contribution to the field of organ-on-chip technology applied to skin tissue engineering. I have no hesitation to recommend publication: the literature search was well conducted, the major critical points were well discussed and the challenges ahead were mentioned as well.
I only suggest to further stress the relevancy and significance of the topic with reference to the general scope of the journal “Pharmaceutics”. The issues related to drug development and testing were – of course - already mentioned, but I suggest to further emphasize these aspects, especially in Abstract and Conclusions.
Overall, a very valuable paper.
Author Response
First, we want to thank the reviewer for its time and for recommending the publication of our manuscript.
Reviewer: I only suggest to further stress the relevancy and significance of the topic with reference to the general scope of the journal “Pharmaceutics”. The issues related to drug development and testing were – of course - already mentioned, but I suggest to further emphasize these aspects, especially in Abstract and Conclusions.
Author response: We agree with the reviewer and added key sentences to the abstract and conclusions. The relevance of the skin-on-a-chip models is discussed in more detail in the introductory chapter. Throughout the text we also try to give examples describing the potential of these models for testing pharmaceutical components. We added some key sentences to emphasize the relevance of the skin-on-a-chip models for drug testing applications. Bellow, we present some examples:
“Abstract: The increased demand for physiologically relevant in vitro human skin models for testing pharmaceutical drugs has led to significant advancements in skin engineering”.
“….Moreover, integrating sensors on the SoC device allows real-time, non-destructive monitoring of skin function and the effect of topically and systemically applied drugs. In this Review, the major challenges and key prerequisites for the creation of physiologically relevant SoC devices for drug testing are considered.
“Conclusions: Although the use of SoC devices to generate biomimetic skin is still in its infancy, much progress has been made in this field. These devices have been used to simulate diseases, bacterial infections and to test drugs regarding toxicity and efficacy. Furthermore, the first steps have been done to establish a co-culture of different tissues to study the systemic effect of topically applied drugs.
“The development of physiological relevant SoC models will be crucial to obtain a useful tool to study the efficacy and toxicity of skin-targeted drugs and to evaluate the systemic effects of pharmaceutical compounds if connected with other OoCs (multi-OoC).”